# Therapeutic Potential of Allicin and Aged Garlic Extract in Alzheimer’s Disease

**DOI:** 10.3390/ijms23136950

**Published:** 2022-06-22

**Authors:** Paola Tedeschi, Manuela Nigro, Alessia Travagli, Martina Catani, Alberto Cavazzini, Stefania Merighi, Stefania Gessi

**Affiliations:** 1Department of Chemical, Pharmaceutical and Agricultural Sciences—DOCPAS, University of Ferrara, 44121 Ferrara, Italy; paola.tedeschi@unife.it (P.T.); martina.catani@unife.it (M.C.); alberto.cavazzini@unife.it (A.C.); 2Department of Translational Medicine, University of Ferrara, 44121 Ferrara, Italy; manuela.nigro@unife.it (M.N.); alessia.travagli@edu.unife.it (A.T.); gss@unife.it (S.G.)

**Keywords:** garlic extracts, Alzheimer’s disease, inflammation, oxidative damage, mechanism of action, NLRP3 inflammasome

## Abstract

Garlic, *Allium sativum*, has long been utilized for a number of medicinal purposes around the world, and its medical benefits have been well documented. The health benefits of garlic likely arise from a wide variety of components, possibly working synergistically. Garlic and garlic extracts, especially aged garlic extracts (AGEs), are rich in bioactive compounds, with potent anti-inflammatory, antioxidant and neuroprotective activities. In light of these effects, garlic and its components have been examined in experimental models of Alzheimer’s disease (AD), the most common form of dementia without therapy, and a growing health concern in aging societies. With the aim of offering an updated overview, this paper reviews the chemical composition, metabolism and bioavailability of garlic bioactive compounds. In addition, it provides an overview of signaling mechanisms triggered by garlic derivatives, with a focus on allicin and AGE, to improve learning and memory.

## 1. Introduction

Botanically, garlic is known as *Allium sativum* L. (family Liliaceae). The exact origin of the name is unknown, but a relation to the Latin word Olere meaning “to smell” is often pointed out. Garlic grows to approximately 30–90 cm in height in well-fertilized, sandy, and loamy soil during spring and summer [1]. For centuries, common garlic cloves have been widely used as food, as well as seasoning for food [2]. The potency of garlic (*Allium sativum*) has been acknowledged for approximately 5000 years. In ancient times, garlic was frequently used as a remedy for intestinal disorders, flatulence, worms, respiratory infections, skin diseases, wounds, symptoms of aging and many other ailments, and abundant literature supports the health-promoting effects of garlic and its by-products, which are associated with the bioactive compounds present in their matrix [3,4,5]. It has been used to reduce triglycerides and low-density cholesterol in the human body [6,7], to lower blood pressure [8], for antithrombotic activity [9], and to increase blood insulin levels [10,11]. In recent years, garlic organosulfur derivatives and garlic supplements have been shown to negatively affect the growth of tumor cells and the risk of cancer [12,13,14,15,16,17,18,19]. Antimicrobial, antiviral, and fungitoxicity activities against different pathogens have also been demonstrated [20,21,22,23,24].

Interestingly, the therapeutic potential of garlic extract in treating Alzheimer’s disease (AD) has been examined in different studies [25]. As a result, the goal of this review was to examine recent advances in the antioxidant and neuroprotective properties of garlic extracts and their main components, focusing on allicin and aged garlic extracts (AGEs), with the goal of reporting what has been learned thus far and identifying a potential role for these agents in the treatment of people suffering from cognitive diseases, such as AD.

## 2. Garlic Bioactive Compounds and Chemistry

### 2.1. Garlic Proximate Composition

A garlic bulb contains approximately 65–68% water; 28% carbohydrates; 2% protein; 1.2% free amino acids; 1.5% fiber, fatty acids, minerals (high levels of potassium, phosphorus, zinc, sulfur, moderate levels of selenium, calcium, magnesium, manganese and iron, and low levels of sodium), phenolic compounds (gallic acid, rutin, quercetin, ferulic acid, p-coumaric acid, naringenin, apigenin, isorhamnetin and luteolin), vitamin A, C and B-complex; and 2.3% organosulfur compounds (Figure 1) [26]. 

### 2.2. Allicin and Organo-Sulfur-Containing Compounds (OSCs)

The numerous biological properties of garlic are mainly attributed to the high contents of bioactive compounds (i.e., organosulfur compounds, phenolic compounds and fructans) [4]. The most significant components, medicinally, are the OSCs (approximately 3–35 mg/g fresh garlic) [27]. OSCs are generally classified into two groups: oil- and water-soluble OSCs (Figure 2). Fresh garlic cloves contain mainly alliin (S-allyl L-cysteine sulfoxide), followed by methiin (S-methylcysteine sulfoxide) and isoalliin, which are formed from the γ -glutamyl cysteine. When garlic is crushed or chewed or cut, alliinase is released, and the conversion of alliin into allicin (allyl 2-propenethiosulfinate) is performed [28]. Therefore, allicin is not found in intact cloves of garlic; both alliin and the enzyme are found in different parts of the bulb or clove [29,30]. Alliin and other cysteine sulfoxides are found in the mesophyll cell, whereas alliinase is localized to a few vascular bundle sheath cells around the veins or phloem. This enzyme is approximately 10 times more abundant in the cloves than in the leaves and represents at least 10% of the total protein in the cloves [31]. The process of allicin production is associated with the defense mechanisms of the plant. Alliinase and alliin form an enzyme-substrate complex in the presence of water, at an optimum temperature of 33 °C and pH of 6.5. Indeed, the enzyme is sensitive to acids, suggesting the employment of enteric-coated formulations of garlic supplements [11,31,32]. Allicin (the most abundant thiosulfinate formed via allinase reactions, approximately 70%) is poorly soluble in water and is responsible for a pungent and unpleasant flavor, but it is very unstable and easily transformed into a high number of products, classified as oil-soluble compounds, mainly including dithiins (formed by the dimerization of thioacrolein created via allicin β-elimination), followed by ajoene, allyl methyl trisulfide (AMTS), diallyl sulfide (DAS), diallyl disulfide (DADS), and diallyl trisulfide (DATS). This breakdown occurs within hours at room temperature and within minutes during cooking [33,34,35,36]. When garlic is extracted in aqueous solvent, γ-glutamyl -S-alk(en)yl-L-cysteines are converted into S-allylcysteine (SAC), S-allylmercaptocysteine (SAMC), metabolites allyl mercaptan (AM) and allyl methyl sulfide (AMS), which are water-soluble organosulfur compounds that are less odorous and more delicate, and have a less characteristic flavor than oil-soluble OSCs. However, although water-soluble OSCs make up a small portion of garlic, they may be considered the main bioactive component in health benefits, such as in cancer prevention and treating AD, as described below [37,38].

### 2.3. Aged Garlic Extract

There is a well-known and interesting product called aged garlic extract (AGE), which warrants special attention. AGE preparation is usually produced by storing sliced garlic cloves in a non-toxic solvent, usually a mixture of water and ethanol (15–20% ethanol solution in water), which is then aged for more than 20 months at room temperature; then, the extract is filtered and concentrated. This aging process aims to transform the odorous and pungent sulfur compounds to odorless ones. This process is characterized by the transformation of allicin into stable and safe sulfur compounds, in particular, water-soluble organosulfur compounds, including the two major ones, SAC and SAMC, and small amounts of oil-soluble allyl sulfides. The composition of AGE garlic extract has many antioxidants properties and, combined with the high bioavailability of SAC and SAMC which are rapidly absorbed by the intestinal tract, seems to play an important role in the biological effects of garlic [13,36,39,40].

## 3. Metabolism and Bioavailability

Although there are many studies regarding the sulfur compounds of garlic and its biological properties, we have little information on the metabolism of garlic and sulfur compounds and related bioavailability [36,41]. The study of bioaccessibility and also bioavailability is important to evaluate the health benefits of bioactive compounds [42,43]. Following the chewing and ingestion of garlic, allicin is synthesized and transformed into its metabolites, which are transported through the stomach, intestine and blood to the target tissues (organs), while losses are channeled through the breath, urine and stool [44]. Therefore, considering the oil- and water-soluble derivatives of garlic from different studies, it is interesting to note that allicin, sulfides, ajoene, vinyldithiins and other oil-soluble OSCs cannot be detected in the blood or urine even after the intake of a large amount of garlic, while after the intake of fresh garlic SAC and water-soluble compounds can be detected in the plasma, liver and kidney [45]. The pharmacokinetics of water-soluble OSCs were found to be different from those of oil-soluble ones. Indeed, SAC was rapidly absorbed in the gastrointestinal tract and was identified in blood and its concentration, and other pharmacokinetic parameters were correlated with SAC doses administered to animals [13,46]. The results from different evaluations on the bioavailability and efficacy of SAC indicated that water-soluble garlic compounds seem to play an important role in the biological effects of garlic, in vitro and in vivo.

## 4. AD

AD represents the most common form of dementia affecting the aging population. It is a progressive neurodegenerative disease affecting memory whose diagnosis is based on cognitive impairment evaluation. Etiology is multifactorial, including both genetic and epigenetic elements. Physical exercise, diet, lifestyle and environmental exposure to heavy metals are crucial factors in its pathogenesis [47,48,49]. From a pathological point of view, AD is characterized by the accumulation of amyloid β-protein (Aβ) plaques and the deposition of hyperphosphorylated tau proteins in the brain, leading to synaptic and neuronal loss [50,51]. Aβ peptide, a 40–42 amino acid residue, is derived from a transmembrane amyloid precursor protein (APP), following cleavage by β- and γ-secretase (BACE1, PS1, PS2 and nicastrin) [52]. After polymerization, the Aβ oligomeric structures devolve into a hazardous molecule, which activates microglia and produces reactive oxygen species (ROS) and inflammatory cytokines, resulting in severe neuronal injury [53,54]. Various amyloid cascade hypothesis-related AD animal models have been developed over the years in order to identify novel therapeutic medications for AD therapy [55,56]. Indeed, it has been shown that the intracerebral administrations of Aβ induces neurodegeneration, as well as deficiencies in learning and memory [55,57]. According to the cholinergic theory, the primary cause of AD is a decrease in acetylcholine (ACh) contents, which are enzymatically destroyed by acetylcholinesterase (AChE) and butyrylcholinesterase (BChE). As a result, these enzymes are important therapeutic strategies for AD. Cholinesterase inhibitors (ChEIs) block these enzymes, resulting in greater ACh levels and a transient alleviation in AD symptoms. Therefore, increasing synaptic ACh concentrations with ChEIs, such as donepezil, rivastigmine and galantamine, is the main current pharmacological therapy for AD patients [58,59]. Unfortunately, these medications, as well as memantine, which decreases glutamate excitotoxicity, only provide symptomatic alleviation and do not slow disease progression; thus, the development of new therapeutic approaches is required. There is currently no conventional medicine for the treatment of AD that is able to modify the pathology. Aducanumab, a novel antibody that targets Aβ protein accumulation, has recently hit the market but its clinical usefulness is still debatable [60,61,62]. In conclusion, as AD is a major health problem, new medications to treat it continue to pique attention.

## 5. Neuroinflammation

Neuroinflammation is implicated in the pathogenesis of neurodegenerative disorders, such as AD [63]. Microglia are immune cells in the brain that react to tissue injury and healing, and their activation contributes to neuroinflammation and neurodegenerative disorders [64,65,66].

Activated microglia spread and replicate, generating pro-inflammatory cytokines including interleukin 1 (IL-1)β and tumor necrosis factor (TNF)α, as well as oxygen and nitrogen radical species, L-glutamate and prostaglandin E2 [67,68]. In addition, other less recognized neurotoxic molecules released by microglia include Aβ, cathepsin B and D, C-X-C motif chemokine ligand (CXCL)10 and CXCL12 (5–67), high mobility group box, lymphotoxin-α, matrix metalloproteinase (MMP)-2 and MMP-9, platelet-activating factor and prolyl endopeptidase, resulting in neuron death or impairment [69].

Multiple signaling pathways responsible for neuroinflammation have been described [70]. Following the identification of invading pathogens and/or tissue damage by pattern recognition receptors (PRR), innate immune activation in the CNS can be initiated through a variety of routes. Considerable emphasis has been placed on a two-signal theory controlled by Toll-like receptors (TLRs) and nucleotide-binding oligomerization domain-containing protein (nucleotide-binding oligomerization domain-containing protein (NOD)-like receptors (NLRs), the latter forming the inflammasome essential for pro-IL-1 and pro-IL-18 digestion [71]. Several TLRs have been shown to interact with pathogen-associated molecular patterns (PAMPs) expressed on broad microbial subclasses, as well as with endogenous molecules, identified as danger-associated molecular patterns (DAMPs), increasing during cell damage [72]. Signal 1 can be triggered by TLR, through the TLR ligand, and the subsequent recruitment of Myeloid Differentiating factor 88 (MyD88) or via the TNF-α receptor, resulting in the NF-κB-dependent transcriptional activation of pro-IL-1 and pro-IL-18 (Figure 3). TLRs and TNF-α are membrane-spanning receptors, but NLRs are intracellular detectors that oligomerize to generate the inflammasome, a multi-protein complex that requires caspase-1 to convert pro-IL-1 and pro-IL-18 to their mature forms. The NOD signaling, accompanied by proteolytic cleavage via inflammasome activation, is essential for this process and represents signal 2, which can be stimulated by a variety of chemicals, comprising different pore-forming toxins, Aβ, ATP, K^+^ outflow, silica and uric acid crystals [73,74,75,76] (Figure 3). NLRP3 is the best-described inflammasome of the 22 NLR genes found in humans, and the observation that multiple structurally diverse stimuli are capable of triggering NLRP3 inflammasome activity has led to the notion that NLRP3 detects a general “danger” signal caused by lysosomal dysfunction [76,77]. Other NLRs, on the contrary, react to a more limited stimulus repertoire. IL-1 and IL-18 have been linked to the pathogenesis of a variety of neurodegenerative illnesses, including AD, as well as a number of CNS pathologies.

Alternatively, TLR can interact with MyD88, triggering NF-κB stimulation, thus provoking the secretion of TNF-α and IL-6 [68,78,79,80].

Other signaling pathways linked to neuroinflammation involve cyclooxygenase 1 (COX1) and 2 (COX-II) activation and the related production of prostaglandins by means of ROS activation, with COX I being generally considered as the housekeeping enzyme and COX II being the enzyme responsible for triggering neuroinflammation [81,82]. 

The PI3K/AKT pathway, including the mammalian target of rapamycin (mTOR), is another important signaling route mediating microglia damage [83,84]. Its activation regulates NF-κB activity in neuroinflammation [85,86]. Furthermore, mitogen-activated protein kinase (MAPK) family kinases, comprising p38 MAPK and stress-activated protein kinases/Jun amino-terminal kinases (SAPK/JNK), play a role in microglia activation, promoting the secretion of proinflammatory cytokines [87]. 

## 6. Garlic Bioactive Compounds as Neuroprotective Agents against AD

### 6.1. Allicin

Recently, the healthy properties of allicin have been widely characterized in terms of neuroprotection for AD therapy [11,88]. A comparison between raw and steamed garlic has been performed in terms of anti-inflammatory activities in LPS-stimulated BV2 microglial cells. It was found that only raw garlic, in which the generation of allicin was preserved, was able to reduce proinflammatory cytokines such as IL-1ꞵ, TNF-α, MCP-1, NO and NF-kB [89]. In particular, an antineuroinflammatory effect was obtained through concentrations of 200 and 400 μM of DATS and DADS, respectively, derived from allicin. Interestingly, the neuroprotective activity of allicin has been investigated in an animal model of AD expressing human double mutant APP and PS1 genes. Allicin induced an improvement in memory by affecting the APP metabolism, with a decrease in Aβ peptide expression, mediated through PS1, PS2, BACE1 and nicastrin reduction. Allicin lowered the elevated levels of oxidative stress (JNK/c-jun-dependent), which are strictly connected to AD pathology, thus reducing mitochondrial dysfunction [90] (Table 1).

Interestingly, it has been reported that JNK/c-jun is associated with APP, Aβ and cognitive decline, suggesting that its blocking may be relevant in the pathogenesis and therapy of AD [40,97,98]. Accordingly, in a previous study, allicin therapy effectively improved age-induced cognitive impairment by increasing the Nrf2 antioxidant signaling pathways, thus protecting the hippocampus against free radical-induced damage [91]. Similarly, in the context of neuroinflammation in depressive-like animal models, allicin reduced microglia activation, and inflammatory cytokines and ROS increased in the hippocampus, while it stimulated SOD activity and Nrf2 signaling. Interestingly, allicin inhibited an overstimulated NLRP3 inflammasome, including a reduction in its components, such as ACS, caspase-1 and IL-1β [92]. Accordingly, in another brain damage-like acute TSCI, causing glial reactivity and neuroinflammation, allicin prevented neuronal injury by limiting oxidative stress, inflammatory response and neuron apoptosis. These events are mediated through Nrf2 pathway activation [93].

The inhibiting action of allicin (IC_50_ = 62 μM) on enzymes that break down acetylcholine, especially AChE and BChE (from bovine erythrocytes), may also play a role in the favorable effect on cognition [99]. In addition, allicin improved cognition and decreased tau phosphorylation and Aβ42 deposit in the hippocampus through the pERK/Nrf2 antioxidative signaling pathway in a model of endoplasmic reticulum stress-related cognitive deficits, obtained through treating rats with TM [94]. Similarly, allicin had a favorable effect on cognition, learning and memory impairment in an AD mouse model. These benefits may be due to an elevation in SOD and a decrease in malondialdehyde activities. Expressions of Aβ and p38 MAPK were also reduced by allicin in these AD mice [95]. Recently, allicin had a protective effect when administered in rats treated with aluminum chloride and copper sulfate as animal models of AD. Specifically, allicin attenuated Aβ plaque formation, oxidative stress, neuroinflammation and cholinergic neuron damage, decreasing the symptoms typical of AD [96]. In conclusion, by engaging numerous molecular and signaling transduction pathways, allicin may be a candidate molecule for combating neuroinflammation (Figure 4) [88].

### 6.2. AGE

There is a large body of evidence dating back to the 1990s that reports on the favorable benefits of AGE on memory and neuroprotection (Figure 5) [25,100,101,102].

In particular, in spontaneous senescent mice, the impairment of learning and memory, as well as brain atrophy, was reduced by AGE [103,104] (Table 2).

In addition, in an in vitro cell model of AD represented by Aβ-injured PC12 cells, AGE and one of its major constituents, SAC, reduced ROS production, caspase-3 activation, DNA fragmentation and PARP cleavage, thus protecting against Aβ-induced apoptosis [105,106,109,121,122,123]. In addition, SAC was protective and reduced cell death in organotypic hippocampal slice culture injured with Aβ [107]. AGE and SAC protected neuronal cells from ROS-mediated damage and preserved the levels of the synaptosomal-associated protein of 25kDa (SNAP25) in the presynaptic neuron [108]. Furthermore, SAC reduced neuronal cell death induced by TM, alone and in combination with Aβ, by blocking calpain-, caspase 12- and caspase 3-dependent pathways [110,111]. Accordingly, AGE was able to improve the cognitive functions in both transgenic and Aβ-treated mice. AGE decreased Aβ deposition, thus hampering hippocampal-dependent memory damage [112,113,114]. Further studies indicate that SAC protects against oxidative damage by restoring the levels of antioxidative enzymes in AD animal models obtained through different approaches, such as D-galactose and STZ mice treatment [115,116]. AGE could alleviate LPS-dependent cognitive deficits via a reduction in oxidative stress, neuroinflammation, astrogliosis, and acetylcholinesterase activity [117]. In addition, AGE can ameliorate working and reference memory, by raising glutamate vesicular transporter 1 protein and glutamate decarboxylase levels and hamper the degeneration of cholinergic neurons [118]. The possible pathway responsible for these protective effects of AGE is related to a reduction in microglia activation in the cerebral cortex and hippocampus of Aβ-induced and transgenic AD animal models, with the consequent inhibition of IL-1β and inflammation [112,119,124]. In addition, AGE impaired cognitive damage induced by scopolamine via a reduction in oxidative stress and AchE activity in mice [120].

Other molecules in AGE, on the other hand, might play a role in neuroinflammation, and the mechanism by which AGE modulates neuroinflammation requires further research.

## 7. Conclusions

Finding a viable molecule for the pharmacological therapy of AD has been one of the most difficult issues in medical research. As most medications aimed at various targets have failed to offer a medical solution, natural products or nutraceutical components, such as garlic, arise as potential protective treatment options. Given that AD is a multifactorial illness, garlic extracts provide the benefit of a multitarget strategy, targeting different biochemical locations in the human brain, as opposed to an individual action, as with most medications used to treat AD. The data given in this review highlight the beneficial antioxidant and neuroprotective anti-inflammatory properties of allicin and AGE contained in garlic extracts. However, these studies are derived from cellular or mouse models, while clinical trials concerning these compounds and AD are not present, suggesting that this evidence should be confirmed in human studies before the beneficial effects of compounds contained in garlic might be translated into therapy. Finally, the importance of the NLRP3 inflammasome pathway in neuroinflammation and neurodegenerative diseases, such as AD, has been established [125], and very recent data on its modulation via allicin have been reported, whereas no evidence has been shown for AGE, implying the need for further characterization.

## Figures and Tables

**Figure 1 ijms-23-06950-f001:**
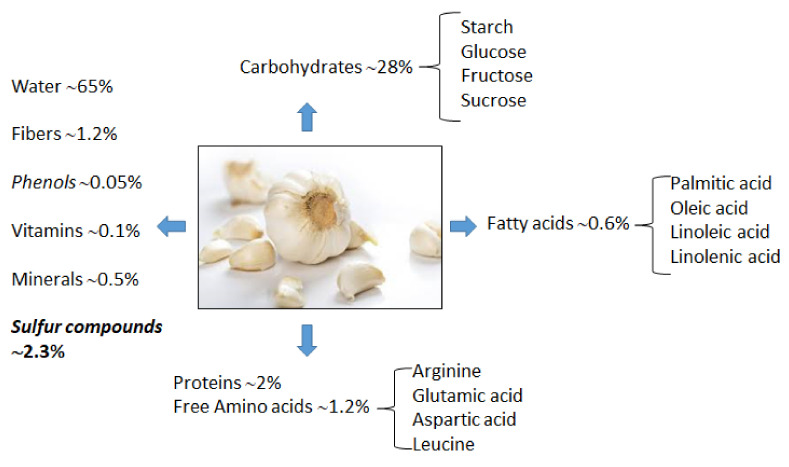
Fresh garlic proximate composition; data are expressed as % *w*/*w*.

**Figure 2 ijms-23-06950-f002:**
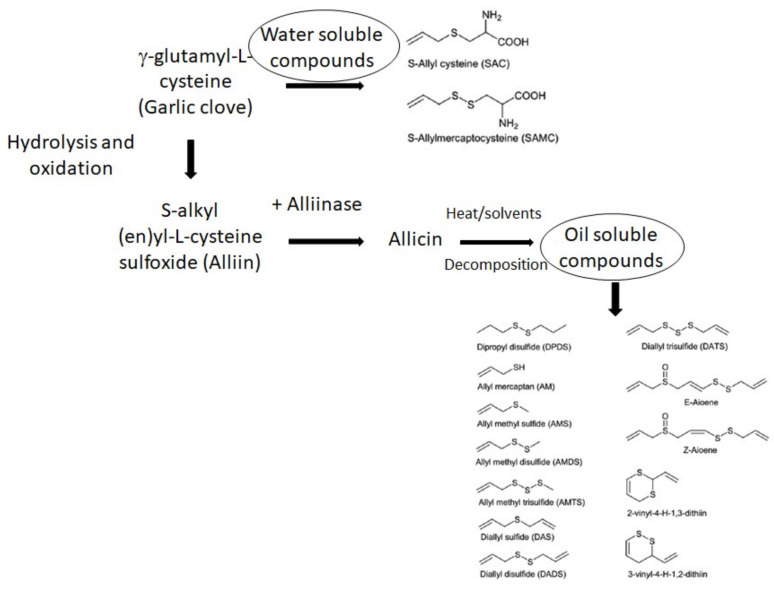
OSCs from garlic. Intact clove garlic contains γ-Glutamyl-S-alk(en)yl-L-cysteines, the primary sulfur compounds, which with hydrolysis or oxidation can be converted into alkyl (en)yl-L-cysteine sulfoxide, commonly called Alliin. When garlic is crushed or cut, the vacuolar enzyme alliinase converts alliin to allicin, poorly soluble in water and responsible for the characteristic pungent flavor of garlic. Allicin is very unstable and rapidly decomposed to form a variety of oil-soluble compounds, including diallyl disulfide (DADS), diallyl sulfide (DAS), diallyl trisulfide (DATS), vinyl dithiin and ajoene, according to different conditions. When garlic is extracted in an aqueous solvent, γ-glutamyl -S-alk(en)yl-L-cysteines can be converted into water-soluble compounds, mainly *S*-allyl-cysteines (SAC) and S-allylmercaptocysteine (SAMC), which are less odorous than the oil-soluble products but are stable and have important antioxidants and bioactive effects.

**Figure 3 ijms-23-06950-f003:**
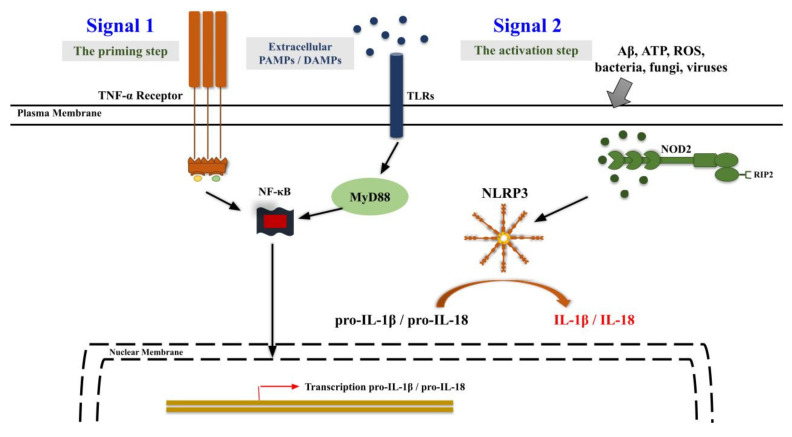
The ways through which the NLRP3 inflammasome is activated: The activation of the NLRP3 inflammasome is a complicated regulatory mechanism that requires two triggering steps: priming and activation. TLR4 or TNF-α receptor agonists cause the priming phase (signal 1), which then activates the NF-B pathway by boosting the production of pro-IL-18 and pro-IL-1. The NLRP3 inflammasome is then stimulated (activation step) by numerous triggering events, such as Aβ, ATP, ROS, bacteria, fungi and viruses (signal 2). Then, caspase-1 can induce maturation of IL-18 and IL-1.

**Figure 4 ijms-23-06950-f004:**
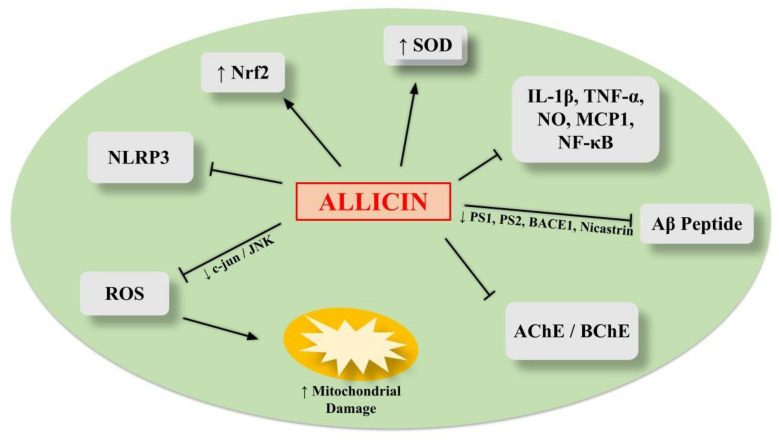
The main events induced by allicin to provide protection in AD.

**Figure 5 ijms-23-06950-f005:**
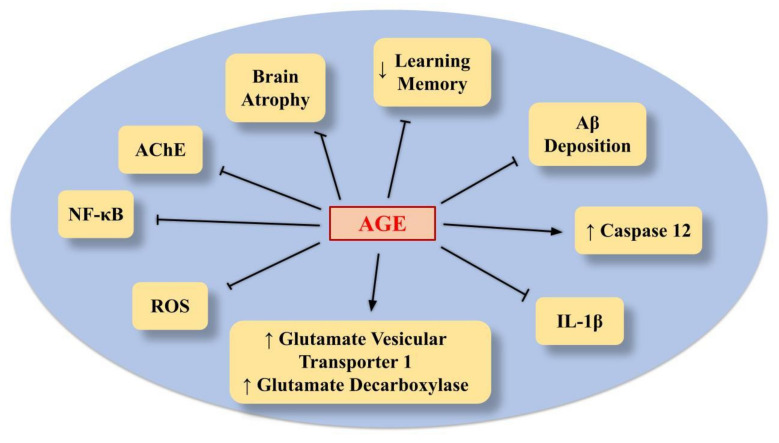
The main events induced by AGE to provide protection in AD.

**Table 1 ijms-23-06950-t001:** Summary of experimental details, including model, chemical concentration and main results of in vitro and in vivo studies of allicin and AD pathology.

References	Experimental Model	Chemicals Concentration	Results
[90]	APP/PS1 double transgenic mice	10 mg/kg/day allicin via intragastric administration on alternate days for 3 months	- Improves cognition - Reduces Aβ expression in the brain - Decreases oxidative stress and improves mitochondrial dysfunction by JNK/c-jun
[91]	3- and 20-month-old C57BL/6 mice	Diet supplemented with 180 mg/kg/day of allicin for 8 consecutive weeks	- Improves age-induced cognitive impairment by increasing the nuclear factor (erythroid-derived 2)-like 2 (Nrf2) transcription factor - Recognizes the human Antioxidant Response Element binding site within glutamylcysteine synthetase, and NADPH:quinone oxidoreductase 1 and defends the cell against free radical-induced damage
[92]	8-week-old male C57BL/6 J mice (depressive-like model)	2, 10, 50 mg/kg allicin once day via intraperitoneal injection	- Decreases ROS production and microglial activation - Upregulates superoxide dismutase (SOD) and Nrf2/HO-1 pathways - Attenuates neuronal apoptosis - Inhibits NLRP3 inflammasome hyperactivity, ACS, caspase-1, and IL-1β proteins
[93]	Rats with acute traumatic spinal cord injury (TSCI)	2, 10, 50 mg/kg intraperitoneal injection of allicin for 21 days	- Induces neuroprotection through antiapoptotic, anti-inflammatory and antioxidant effects
[94]	Adult male rats 72 h of lateral ventricular infusion of tunicamycin (TM), an endoplasmic reticulum stress stimulator (cognitive deficit induction)	Diet supplemented with 180 mg/kg/d of allicin for 16 weeks	- Decreases tau phosphorylation and Aβ42 deposit in the hippocampus, oxidative stress - Increases pERK and Nrf2 expression in the hippocampi
[95]	AD mouse model via injection of Aß(1–42) (1 µL = 4 µg) into the bilateral hippocampi	Allicin via intraperitoneal injection for 14 days 180 mg/kg/day)	- Prevents learning and memory impairment - Increases SOD and decreases ROS
[96]	Male Wistar rats	Allicin via intraperitoneal injection (10 and 20 mg/kg) 7 days before metals (aluminum chloride, 200 mg/kg p.o; copper sulfate, 0.5 mg/kg p.o.) administration for 28 days	- Exhibits neuroprotective effect through antioxidant, anti-inflammatory, neurotransmitter restoration; attenuation of neuroinflammation; and β-amyloid-induced neurotoxicity

**Table 2 ijms-23-06950-t002:** Summary of experimental details, including model, chemical concentration and main results, of in vitro and in vivo studies of AGE and AD pathology.

References	Experimental Model	Chemicals Concentration	Results
[104]	Senescence-Accelerated Mice (SAM)	Diet containing 2% (*w*/*w*) AGE	- Provides antiaging effect, increases the survival ratio and ameliorates the memory acquisition deficit and the memory retention impairment
[105]	Neuronal PC12 cells treated with NGF for 4 days and injured with Aβ 95 nM	Cell growth medium containing 0.01% AGE	- Protects neuronal PC12 cells against Aβ toxicity
[106]	Undifferentiated PC12 cells injured with Aβ25-35 40 μM	AGE (1–8 mg/mL) SAC (1–4 mg/mL)	- Reduces ROS and apoptosis
[107]	Hippocampal slice culture injured with Aβ25–35 25 μM	SAC (10–100 μM)	- Protects from cell death induced by Aβ25–35
[108]	Neuronal PC12 cells treated with NGF for 12 days and APP-Tg mice	0.3% or 1.0% AGE in PC12 cells diet 2% AGE in mice	- Protects from ROS-mediated damage
[109]	NGF-treated neuronal PC12 cells injured with Aβ25-35 80 μM ICR mice administered Aβ25-35 via intracerebroventricular injection	AGE 25–200 µg/mL in PC12 cells Freeze-dried ethyl acetate fraction from AGE at concentrations of 5, 10 and 20 mg/kg in mice	- Decreases in vitro ROS accumulation - Improves cognitive impairment against Aβ-induced neuronal deficit
[110]	rat hippocampal neurons injured with Aβ25–35 5 μM or TM (10 μg/mL)	SAC (1 µM)	- Decreases neuronal cell death, ROS and caspase 12 induced by Aβ25–35 or TM
[111]	rat hippocampal neurons injured with Aβ25–35 25 μM or TM (20–80 μg/mL)	SAC (100 µM)	- Blocks Aβ potentiation of TM neurotoxicity - Reverses the increase in calpain activity and the active forms of caspase-12 and caspase-3 induced by Aβ + TM
[112]	Tg2576 mice	AGE (40 mg/kg/d/4 wks)	- Increases sAPPalpha - Decreases Aβ40 and Aβ42 deposition
[113]	Tg2576 mice	AGE 2%, 20 mg SAC/kg and 20 mg DADS/kg	- Decreases cerebral plaque, inflammation and TAU-GSK3ꞵ-dependent phosphorylation. The order of ameliorative efficacy - is AGE > SAC > DADS
[114]	Tg2576 mice	Diet 2% AGE for 5 months	- Prevents deterioration of hippocampal-based memory tasks
[115]	C57BL/6 mice treated with D-galactose (AD-like model)	SAC (1 g/L into drinking water for 7 weeks)	- Decreases the production of Aβ and suppresses the expression of APP and BACE1 - Retaines PKC activity, and the expression of PKC-α and PKC-γ - Decreases ROS and protein carbonyl levels and restores brain GPX, SOD and catalase activities - Lowers aldose reductase (AR) activity, AR expression, and carboxymethyllysine and pentosidine levels
[116]	Intracerebroventricular infusion of streptozotocin (STZ) (model of memory impairment in mice)	SAC (30 mg/kg i.p. for 15 days)	- Prevents increased latency and path length and attenuates oxidative stress induced by STZ
[117]	LPS-treated rats (167 μg/kg for 7 days)	SAC (25, 50, 100 mg/kg/day p.o. for 7 days)	- Increases cognition, learning and memory reduced by LPS; increases SOD and GSH - Reduces acetylcholinesterase activity NF-κB, TLR4, GFAP and IL-1β, and increases Nrf2
[118]	Rats injured with Aβ1–42 1 µg/µL intracerebroventricular infusion	AGE (125, 250 and 500 mg/kg p.o. for 65 days)	- Ameliorates working and reference memory by raising glutamate vesicular transporter 1 protein and glutamate decarboxylase levels - Restores cholinergic neuron density reduced by Aβ1–42
[119]	Rats injured with Aβ1–42 1 µg/µL intracerebroventricular infusion	AGE (125, 250 and 500 mg/kg body weight, p.o., daily for 56 days)	- Improves short-term recognition memory in cognitively impaired rats - Reduces microglial activation and IL-1β
[120]	Scopolamine-treated mice (2 mg/kg) injected 30 min before the tests.	AGE (25 or 50 mg/kg p.o.)	- Protects against scopolamine-induced cognitive impairment by decreasing oxidative damage and regulating cholinergic function - Increases levels of glutathione, glutathione peroxidase and glutathione reductase, and inhibits lipid peroxidation - Attenuates cholinergic degradation by inhibiting acetylcholinesterase activity and increasing choline acetyltransferase activity

## Data Availability

Not applicable.

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
