# Peer review of "Therapeutic Potential of Allicin and Aged Garlic Extract in Alzheimer’s Disease"

_ijms, 2022, doi:10.3390/ijms23136950_

Round 1
Reviewer 1 Report
Main Points:
- The manuscript has more than 33% plagiarism, normally, the articles less than 20 or 25% are considered for publication in quality journals like IJMS.
- There is a lot of repetition in the text. The manuscript is not well targeted.
- Many components of the garlic have been discussed in the present review. You must have selected a single substance or a group of substances that have been active against AD? Most of the garlic associated bioactive compounds have been individually characterized for their pharmaceutical roles. The sulphur containing bioactive compounds are being sold by Sigma, and crude extract of garlic is now a matter of recent past.
- The sulphur compounds such as Allicin, alliin have already been studied for their antimicrobial, antioxidant, anti-inflammatory and neuroprotective effects. Why you did not select a particular bioactive compound from garlic?
- Phenolic compounds are commonly found in many plants as secondary metabolites, why did you particularly discuss phenolic compounds from garlic? As you have not discussed the role of phenolic compounds, vitamins and minerals in the management of AD, what was the point in a description of such compounds, only one paragraph could be enough.
- The role of Allicin has been recently reviewed in details (https://doi.org/10.3390/antiox11010087). What new information your review has added to the already published data about the role of Allicin in AD management?
- What is the main difference between the composition fresh and aged garlic extract that the authors think has made AGE more potent against AD?
- There is no clinical study reported in the present review describing the role of Allicin and AGE, particularly with reference to AD.
- I recommend a restructuring and rewriting of review article before submission. Also, I suggest to reduce the plagiarism and resubmit a new version of review.

Author Response
Reviewer 1
Main Points:
- The manuscript has more than 33% plagiarism, normally, the articles less than 20 or 25% are considered for publication in quality journals like IJMS.
We thank the reviewer for his/her comments, accordingly we have corrected.
- There is a lot of repetition in the text. The manuscript is not well targeted.
The referee is correct and by considering all the suggestions made by him/her we have re-written the manuscript and sent it for english correction. We now hope that, after these substantial changes, it may be suitable for publication on IJMS.
- Many components of the garlic have been discussed in the present review. You must have selected a single substance or a group of substances that have been active against AD? Most of the garlic associated bioactive compounds have been individually characterized for their pharmaceutical roles. The sulphur containing bioactive compounds are being sold by Sigma, and crude extract of garlic is now a matter of recent past.
We thank the reviewer for his/her comments. The reviewer is correct, therefore, instead of fresh garlic extract, we opted to focus on allicin and AGE, particularly rich in water-soluble chemicals, that are interesting and crucial in AD.
- The sulphur compounds such as Allicin, alliin have already been studied for their antimicrobial, antioxidant, anti-inflammatory and neuroprotective effects. Why you did not select a particular bioactive compound from garlic?·
We thank the reviewer for his/her comments. We chose to focus our work on allicin as well as on AGE, which appear to be linked to important biological activities in neurodegenerative disorders, such as Alzheimer’s disease.
- Phenolic compounds are commonly found in many plants as secondary metabolites, why did you particularly discuss phenolic compounds from garlic? As you have not discussed the role of phenolic compounds, vitamins and minerals in the management of AD, what was the point in a description of such compounds, only one paragraph could be enough.
We thank the reviewer for his/her comments, to better target the manuscript, we focalized the review on garlic sulfur compounds, mainly correlated with the numerous beneficial activities of this vegetable.
- The role of Allicin has been recently reviewed in details (https://doi.org/10.3390/antiox11010087). What new information your review has added to the already published data about the role of Allicin in AD management?
The reviewer is correct. The role of Allicin has been recently reviewed in details in https://doi.org/10.3390/antiox11010087. However, in our manuscript we have also addressed the role of Allicin in NLRP3 inflammasome activation. Inflammasome has recently been identified as a strong target for neurodegenerative diseases, and may offer a novel strategy for neurodegenerative diseases that are currently without effective treatments. In addition, we have also reported the evidence concerning the beneficial effects of AGE in AD.
- What is the main difference between the composition fresh and aged garlic extract that the authors think has made AGE more potent against AD?
In AGE are present organosulfur compounds which are absent in fresh garlic. Now, in the revised version of the review, a new paragraph 2.3 reporting how AGE are obtained was added.
-There is no clinical study reported in the present review describing the role of Allicin and AGE, particularly with reference to AD.
The referee is correct but we have not found clinical trials with allicin or AGE in AD and we have added a sentence at the end of the conclusions.
- I recommend a restructuring and rewriting of review article before submission. Also, I suggest to reduce the plagiarism and resubmit a new version of review.
We thank the reviewer for his/her suggestion that allows us to improve our manuscript.
Reviewer 2 Report
Tedeschi et al. discuss the therapeutic potentials of garlic in the context of Alzheimer disease. The manuscript, as presented, is not befitting peer-review or publication because it is written as a premature draft without deep thinking, and it is not fit for any academic consumption. I suggest the authors seek help from an academic English-editing service provider before resubmitting the text. I have added my comments below and hope that they authors will find them useful. These comments are by no means comprehensive, but only exemplary. The issues are too many to be listed in a peer-review like this considering the scope and time of a referee.
- Scientific conventions: Please note that the WHO-designated nomenclature for Aβ is “amyloid β-protein”. Please use this throughout the manuscript. “Aβ” as an abbreviation is fine. See lines 195, 251, 334.
- Wrong conventions: Despite being the commonly used and commonly accepted “misnomer”, the eponymous term “Alzheimer’s Disease” (AD) is logically incorrect (see the title, lines 15, 21, 54, 191, 192). Historically, the disease was discovered by Alois Alzheimer; the disease was not “his own” personal disease. Because of the eponymous convention, using the possessive form (apostrophe plus “s” or genitive “s”) is wrong but has become perpetual in the English Scientific literature by our great peers. Many though have avoided it (see the titles of references 84 and 88 of this review). The Australian Manual of Scientific Style and The Chicago Manual of Style advise against the use of the possessive I suggest taking their editorial advice and applying it throughout the text.
- Lack of detail or vague expressions: The review is without proper consideration of scientific/academic writing/reporting. It suffers from vague and unclear statements that do not help the reader. See line 53, “against different pathogens”. Line 55, “in different studies”. Line 121, “nothing less”. Line 152, “The results from different evaluations on bioavailability and efficacy of SAC indicated”. Line 160, “a great number of clinical trials”. Line 231, “poisoning molecules”. Line 310, “administering a preventive dosage of allicin had a favourable effect on cognition in an AD mice model”. What is the value of information to the reader in this style of reporting? The reader may ask, “what is a preventive dosage”? “What is a favourable outcome”? What was the AD mouse model”? “What are poisoning molecules”? What does it all mean?
- Lack of relevant and valuable detail: When reporting the scientific literature, please add details of the studies used. For example, it will be informative for the reader to know the aim of a reported study; the model used; if cell model, what cell type; if animal model, what animal; what concentration of what chemical was used; and what the findings or outcomes were; what do these outcomes really mean in the context of human disease? Tabulate all the findings in well-designed tables to summarise the findings reported from the literature.
- Factual inaccuracy: the authors have listed vitamins against their mentioning “trace elements”. See line 64, “trace elements (vitamin C, B1, B6)”. Revise all the statements in the review for factual accuracy.
- English conventions: Avoid the use of noun trains, long strings of nouns acting as adjectives. For example see line 154, “benefits of bioactive compounds rich diet”.
- English conventions/notations: Use the decimal point as used conventionally in the English literature, not European convention.
- What do the reported percentages on page 2 mean? Are these mass percentages? Please add details to make valuable sense.
- Unclear statements: Many statements need to be rewritten either to make sense or improve English. For example, see the statement in line 169, “To better preserve …”. In that statement, “better” is redundant when “preserve” is used. Revise for logic and remove redundancies. See the statement in line 174, “are mainly present …”
- Flawed logic: Revise the logic of the cause-and-effect relationship in the statement starting in line 136. Why is bioavailability an important concept because no full amounts of …? Simply define bioavailability. What does “no full amounts” mean? Use simple and clear language.
- Lack of detail: The diagram in Figure 2 is valuable. The caption should be revised to explain the diagram.
- Bias: The review is biased. For a single example, quercetin is a pan-assay-interference compound (PAINS). Discuss the literature relevant to all the components of garlic, and if any component is identified or evidenced as a PAINS, add that evidence to this review and discuss such evidence. Educate the reader. This will remove the bias in the review and provide an encompassing and informative literature review to the audience. The same applies to certain flavonoids. Without such discussions and evidence, the review does not add any value to the existing literature. No point blindly summarising the literature. See the literature in the context of curcumin acting as a PAINS, for example.
- Avoid stating the obvious/remove redundancies: Do not add the word “enzyme” after mentioning the scientific name of an enzyme. See line 213, “cholinesterase enzymes”. Apply throughout the manuscript in similar occurrences.
- Number agreement: In line 264, plurals are mentioned apart from “Virus”. Capitalisation is not necessary, and “Virus” needs to be plural. The number agreement should be revised at different occurrences throughout the manuscript.
- Flawed logic: in line 291 “mitochondrial dysfunction improvement” means aggravation of the dysfunction. How can a negative concept be improved? How can dysfunction be improved? The logic is flawed. Revise.
- Avoid nominalisations: “provided inhibition” should simply be “inhibited”. See line 301. When a verb is available, do not use nominalization. Revise throughout the manuscript.
- Bias: The review is biased. Discuss the limitations of the cellular or mouse models in the context of the real human disease or the understanding thereof. Without discussing or highlighting the design limitations of such studies, the reader will not understand that the observed or reported therapeutic effects may have been exaggerated.
- The abbreviation “AGE” also refers to “advanced glycation end-products”. Revise the use of proper abbreviations or remove excessive abbreviations.
- Revise punctuation, word choice, and spelling, throughout the text.
Author Response
Reviewer 2
Tedeschi et al. discuss the therapeutic potentials of garlic in the context of Alzheimer disease. The manuscript, as presented, is not befitting peer-review or publication because it is written as a premature draft without deep thinking, and it is not fit for any academic consumption. I suggest the authors seek help from an academic English-editing service provider before resubmitting the text. I have added my comments below and hope that they authors will find them useful. These comments are by no means comprehensive, but only exemplary. The issues are too many to be listed in a peer-review like this considering the scope and time of a referee.
- Scientific conventions: Please note that the WHO-designated nomenclature for Aβ is “amyloid β-protein”. Please use this throughout the manuscript. “Aβ” as an abbreviation is fine. See lines 195, 251, 334.
We have corrected accordingly
- Wrong conventions: Despite being the commonly used and commonly accepted “misnomer”, the eponymous term “Alzheimer’s Disease” (AD) is logically incorrect (see the title, lines 15, 21, 54, 191, 192). Historically, the disease was discovered by Alois Alzheimer; the disease was not “his own” personal disease. Because of the eponymous convention, using the possessive form (apostrophe plus “s” or genitive “s”) is wrong but has become perpetual in the English Scientific literature by our great peers. Many though have avoided it (see the titles of references 84 and 88 of this review). The Australian Manual of Scientific Styleand The Chicago Manual of Style advise against the use of the possessive I suggest taking their editorial advice and applying it throughout the text.
We have corrected accordingly
- Lack of detail or vague expressions: The review is without proper consideration of scientific/academic writing/reporting. It suffers from vague and unclear statements that do not help the reader. See line 53, “against different pathogens”. Line 55, “in different studies”. Line 121, “nothing less”. Line 152, “The results from different evaluations on bioavailability and efficacy of SAC indicated”. Line 160, “a great number of clinical trials”. Line 231, “poisoning molecules”. Line 310, “administering a preventive dosage of allicin had a favourable effect on cognition in an AD mice model”. What is the value of information to the reader in this style of reporting? The reader may ask, “what is a preventive dosage”? “What is a favourable outcome”? What was the AD mouse model”? “What are poisoning molecules”? What does it all mean?
We thank the reviewer for his/her comments. Accordingly, we have provided the experimental requests raised by the reviewer. In addition, we have added experimental details on the study, as requested by the reviewer, adding two tables.
Lack of relevant and valuable detail: When reporting the scientific literature, please add details of the studies used. For example, it will be informative for the reader to know the aim of a reported study; the model used; if cell model, what cell type; if animal model, what animal; what concentration of what chemical was used; and what the findings or outcomes were; what do these outcomes really mean in the context of human disease? Tabulate all the findings in well-designed tables to summarise the findings reported from the literature.
We thank the reviewer for his/her comments. Accordingly, we have added relevant and valuable details by adding a table for allicin and another for AGE.
- Factual inaccuracy: the authors have listed vitamins against their mentioning “trace elements”. See line 64, “trace elements (vitamin C, B1, B6)”. Revise all the statements in the review for factual accuracy.
We thank the reviewer for his/her comments. Accordingly, we have corrected.
- English conventions: Avoid the use of noun trains, long strings of nouns acting as adjectives. For example see line 154, “benefits of bioactive compounds rich diet”.
We have corrected, accordingly.
- English conventions/notations: Use the decimal point as used conventionally in the English literature, not European convention.
We have corrected accordingly.
- What do the reported percentages on page 2 mean? Are these mass percentages? Please add details to make valuable sense.
We thank the reviewer for his/her comments. Accordingly, we have corrected and added the details (the data are expressed as mass percentages).
- Unclear statements: Many statements need to be rewritten either to make sense or improve English. For example, see the statement in line 169, “To better preserve …”. In that statement, “better” is redundant when “preserve” is used. Revise for logic and remove redundancies. See the statement in line 174, “are mainly present …”
We have corrected accordingly.
- Flawed logic: Revise the logic of the cause-and-effect relationship in the statement starting in line 136. Why is bioavailability an important concept because no full amounts of …? Simply define bioavailability. What does “no full amounts” mean? Use simple and clear language.
We thank the reviewer for his/her comments. Accordingly, we have simplified and clarified the concept.
- Lack of detail: The diagram in Figure 2 is valuable. The caption should be revised to explain the diagram.
We thank the reviewer for his/her comments, Accordingly, we have better explained the diagram 2, in order to make understand the transformation and production of important sulfur compounds.
- Bias: The review is biased. For a single example, quercetin is a pan-assay-interference compound (PAINS). Discuss the literature relevant to all the components of garlic, and if any component is identified or evidenced as a PAINS, add that evidence to this review and discuss such evidence. Educate the reader. This will remove the bias in the review and provide an encompassing and informative literature review to the audience. The same applies to certain flavonoids. Without such discussions and evidence, the review does not add any value to the existing literature. No point blindly summarising the literature. See the literature in the context of curcumin acting as a PAINS, for example.
We thank the reviewer for his/her issue. Following the revision we have addressed our work only on the role of allicin and AGE in AD pathology and these compounds do not seem to be included in pan-assay-interference compound.
- Avoid stating the obvious/remove redundancies: Do not add the word “enzyme” after mentioning the scientific name of an enzyme. See line 213, “cholinesterase enzymes”. Apply throughout the manuscript in similar occurrences.
We have corrected accordingly
- Number agreement: In line 264, plurals are mentioned apart from “Virus”. Capitalisation is not necessary, and “Virus” needs to be plural. The number agreement should be revised at different occurrences throughout the manuscript.
We have corrected accordingly
- Flawed logic: in line 291 “mitochondrial dysfunction improvement” means aggravation of the dysfunction. How can a negative concept be improved? How can dysfunction be improved? The logic is flawed. Revise.
Done
- Avoid nominalisations: “provided inhibition” should simply be “inhibited”. See line 301. When a verb is available, do not use nominalization. Revise throughout the manuscript.
Done, the review has been sent to MDPI editing service for english correction
- Bias: The review is biased. Discuss the limitations of the cellular or mouse models in the context of the real human disease or the understanding thereof. Without discussing or highlighting the design limitations of such studies, the reader will not understand that the observed or reported therapeutic effects may have been exaggerated.
The reviewer is correct and we have added a sentence explaining this bias at the end of the conclusions.
18.The abbreviation “AGE” also refers to “advanced glycation end-products”. Revise the use of proper abbreviations or remove excessive abbreviations.
The Reviewer is correct, however the abbreviation “AGE” for “Aged Garlic Extract” is an established abbreviation already present in literature.
- Revise punctuation, word choice, and spelling, throughout the text.
Done. We have used the editing service available to publish with MDPI for english correction
Round 2
Reviewer 2 Report
I thank the authors for revising the text according to most of my recommendations. Some recommendations were ignored despite the fact that the authors mentioned that they have corrected them. For example, using some noun trains remain. A relative phrase using prepositions will avoid using noun trains and clarify the text, making it easy to read. I hope that the journal English editors and typesetters take heed. Some typesetting issues remain in the text, for example, the wrong usage of hyphens instead of en-dashes. Again, these could be identified and corrected during production, hopefully.